# Plasmons in the Kagome metal $CsV_3Sb_5$

H. Shiravi [1,2], A. Gupta[1,2], B. R. Ortiz [3,4], S. Cui[1,2], B. Yu[5], E. Uykur [6,7], A. A. Tsirlin [8], S. D. Wilson [3], Z. Sun [5] ✉ & G. X. Ni [1,2] ✉

Plasmon polaritons, or plasmons, are coupled oscillations of electrons and electromagnetic fields that can confine the latter into deeply subwavelength scales, enabling novel polaritonic devices. While plasmons have been extensively studied in normal metals or semimetals, they remain largely unexplored in correlated materials. In this paper, we report infrared (IR) nano-imaging of thin flakes of $CsV_3Sb_5$, a prototypical layered Kagome metal. We observe propagating plasmon waves in real-space with wavelengths tunable by the flake thickness. From their frequency-momentum dispersion, we infer the out-of-plane dielectric function $\epsilon_c$ that is generally difficult to obtain in conventional far-field optics, and elucidate signatures of electronic correlations when compared to density functional theory (DFT). We propose correlation effects might have switched the real part of $\epsilon_c$ from negative to positive values over a wide range of middle-IR frequencies, transforming the surface plasmons into hyperbolic bulk plasmons, and have dramatically suppressed their dissipation.

Layered metals possess a range of optical properties that are essential for a multitude of applications in photonics, optoelectronics and plasmonics[1–5]. In severe circumstances, their in-plane and out-of-plane dielectric functions display opposite signs, classifying them as hyperbolic materials[6,7] in which light and matter hybridize to generate polaritonic modes known as hyperbolic polaritons[8–11]. These modes feature hyperboloid-shaped iso-frequency surfaces (IFS) in momentum space[12], leading to a plethora of exceptional nano-optical properties[6,7], including unidirectional waveguiding[9,10], Purcell enhancement[13,14], negative refraction[15], subwavelength imaging[16,17] and cloaking[18], among others. Natural hyperbolic materials can sustain a range of hyperbolic frequencies and momenta, allowing them to accommodate the aforementioned phenomena without the extensive fabrication processes required for artificially engineered hyperbolic metamaterials[6].

Systematic investigations have been conducted on natural layered two-dimensional (2D) hyperbolic insulators[19,20] such as hexagonal boron nitride[21–24], $\alpha$-$MoO_3$[25–27], $\alpha$-$V_2O_5$[28], and hyperbolic metals such as $ZrSiSe$[29] and $WTe_2$[30]. However, the effects of electronic correlations on hyperbolic plasmon polaritons in layered metals remain largely unexplored. In addition, hyperbolic plasmons in correlated metals and

superconductors[5,31–33] also offer valuable insights into their many-body physics, which are difficult to examine via conventional optical approaches. Meanwhile, the significant electronic losses typically encountered in layered 2D metals have hindered the experimental observations of hyperbolic plasmons.

The recently discovered Kagome metal family $AV_3Sb_5$ (A = K, Rb or Cs) offers a new platform for exotic physics owing to the interplay of its nontrivial band topology, geometric frustration and electronic correlations[34–38]. The in-plane network of corner-sharing triangles (Fig. 1a) formed by vanadium has triggered various correlation-driven states including charge density waves and unconventional superconductivity[39–41]. Evidence for correlated electrons has been found in both spectroscopic and local density of state probe measurements[40–43]. However, collective electronic excitations in this class of Kagome metals remain largely unknown.

## Results

### Nano-IR probing of thin $CsV_3Sb_5$ crystal

We report the direct observation of plasmons in thin flakes of $CsV_3Sb_5$ using scattering-type scanning near-field optical microscopy (s-SNOM),

[1]Department of Physics, Florida State University, Tallahassee, FL 32306, USA. [2]National High Magnetic Field Laboratory, Tallahassee, FL 32310, USA. [3]Materials Department, University of California Santa Barbara, Santa Barbara, CA 93106, USA. [4]Materials Science and Technology Division, Oak Ridge National Laboratory, Oak Ridge, TN 37831, USA. [5]State Key Laboratory of Low-Dimensional Quantum Physics and Department of Physics, Tsinghua University, 100084 Beijing, China. [6]Physikalisches Institut, Universität Stuttgart, 70569 Stuttgart, Germany. [7]Helmholtz-Zentrum Dresden-Rossendorf, Institute of Ion Beam Physics and Materials Research, 01328 Dresden, Germany. [8]Felix Bloch Institute for Solid-State Physics, Leipzig University, 04103 Leipzig, Germany. ✉e-mail: zysun@tsinghua.edu.cn; guangxin.ni@magnet.fsu.edu

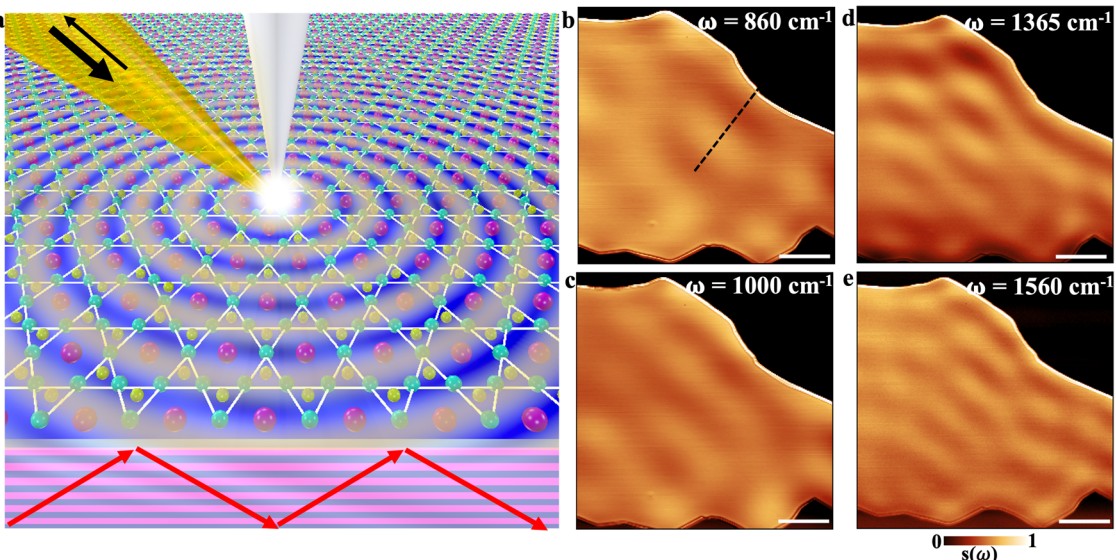

**Fig. 1 | Nano-IR probing of CsV₃Sb₅ crystal. a** Schematic of near-field imaging of the Kagome metal with propagating hyperbolic plasmon waves shown on the top surface (ab-plane) and the cross-section (ac-plane). **b–e** The near-field amplitude $s(\omega)$ plotted in real-space at selected IR frequencies on a 80-nm-thick flake, see details in Supplementary Note 1. The scale bar is 5 μm in length.

which enables local optical excitations of the electronic states at selected frequencies (see Methods). We show that CsV₃Sb₅ hosts propagating plasmons that are highly tunable across a wide range of IR wavelengths spreading beyond $\lambda_{IR} \sim 5.8$–$11.6\,\mu$m (or frequencies $\omega \sim 860$–$1700\,$cm$^{-1}$). The in-plane dielectric function $\epsilon_{ab}$ of CsV₃Sb₅, as determined from previous far-field measurements, resembles that of a Drude metal with a characteristic plasma frequency of about $8000\,$cm$^{-1}$[43,44]. While the in-plane response has been well-characterized, the out-of-plane response $\epsilon_c$ is expected to provide superior insights into correlation effects, remains inaccessible to far-field optics. Through our analysis of plasmonic fringes in real-space, we have deduced the existence of hyperbolic plasmons propagating within the thin flakes of CsV₃Sb₅. Our observation indicates a positive real part of $\epsilon_c$ in the relevant frequency range, contradicting the negative $\epsilon_c$ predicted by the density functional theory (DFT). The sign change, along with the reduced dissipation (imaginary part) of $\epsilon_c$, makes CsV₃Sb₅ a promising platform for investigating the nature of hyperbolic plasmons facilitated by electronic correlations in layered metals.

To access the Kagome plasmons in CsV₃Sb₅, we conducted nano-IR imaging using s-SNOM based on a tapping-mode atomic force microscope (AFM) equipped with a sharp metallic tip (Fig. 1a) (see Methods and Supplementary Note 1). The AFM tip was illuminated by IR light with tunable frequency $\omega = 2\pi/\lambda_{IR}$, generating a strongly enhanced local electric field underneath. This setup resolves the problem of photon-plasmon momentum mismatch[29,33,45–48], enabling the launching of plasmonic waves with wavelength $\lambda_p < \lambda_{IR}$. The local electric field of the plasmonic waves gets scattered into the far-field by the sample edge, facilitating the direct measurement of the plasmonic response with ∼20 nm spatial resolution. We note that the reversed path also contributes to the near-field signal: plasmonic waves launched by the edge, picked up by the tip, and then scattered into far-field (see Supplementary Note 2).

Representative nano-IR imaging data is depicted in Fig. 1b–e, where we show the normalized near-field amplitude $s(\omega)$ at selected excitation frequencies on a 80-nm-thick CsV₃Sb₅ flake. The plasmonic waves form clear interference patterns (fringes) parallel to the edges, whose period decreases systematically with increasing frequency. These findings are consistent with expectations for plasmonic modes dispersing with positive group velocity. Additionally, the fringes extend over the entire sample surface across a wide frequency range,

indicating the long lifetimes of the observed Kagome plasmons. Similar fringe patterns were observed in dozens of CsV₃Sb₅ samples with different thicknesses, including samples as thin as tens of nanometers.

**Frequency-momentum dispersion of thin CsV₃Sb₅**
Nano-imaging results in Fig. 1 allow us to determine the polariton wavelength $\lambda_p$ and the corresponding momentum $q_p = 2\pi/\lambda_p$. In Fig. 2a we show averaged line-profiles at several IR frequencies. To accurately determine $\lambda_p$, we Fourier transformed these profiles to obtain the wave-vector $k^*$. The near-field momentum $q_p = 2\pi/l$ is then extracted by subtracting the 'far-field factor' using the relation $k^* = q_p + k_0 \sin\alpha\cos(\phi - \phi_{plasmon})$, where $\alpha = 60^\circ$ is the incident angle, $\phi = 74.8^\circ$ is the azimuthal angle of the incident beam, and $\phi_{plasmon}$ is the in-plane angle of plasmon propagation[46]. There are typically two potential sources for the near-field period $l$: plasmons traveling a one-way trip between the tip and the edge ($l = \lambda_p$), and plasmons traveling a round trip between the tip and the edge ($l = \lambda_p/2$)[49]. To resolve this uncertainty, we performed nano-IR imaging studies on CsV₃Sb₅ flakes with pre-patterned micron-sized metallic structures on top, which are known to efficiently launch plasmon waves with periodicity of $l = \lambda_p$[24,46]. We observed that the pre-patterned metallic launchers act as fixed plasmonic antennas, producing fringes with the same period as the edges. Furthermore, we found that in thicker flakes, the value of $2l$ is systematically larger than $\lambda_{IR}$ (see Supplementary Note 2). Therefore, we conclude that the observed fringes originate from the one-way propagation of plasmons, where $l = \lambda_p$ (see Supplementary Note 3).

By analyzing the nano-IR data at various laser energies, we construct the frequency-momentum ($\omega, q$) dispersion of these modes and overlay the data on top of the calculated imaginary part of the reflectivity coefficient, $r_p = r_p(\omega, q)$ (Fig. 2b). This colormap of Im[$r_p$] provides an intuitive way to visualize the collective dispersion and damping rate of the plasmonic modes. Specifically, the Im[$r_p$] shown in Fig. 2b is calculated for an 80-nm-thick crystal of CsV₃Sb₅ on a SiO₂/Si substrate using the experimentally obtained in-plane dielectric function $\epsilon_{ab}(\omega)$[43], and the out-of-plane dielectric function $\epsilon_c(\omega)$ from DFT calculations. Clearly, the calculated Im[$r_p$] does not match the measured plasmon dispersion. According to the negative $\epsilon_c$ predicted by DFT, the expected polariton modes would be two surface plasmon

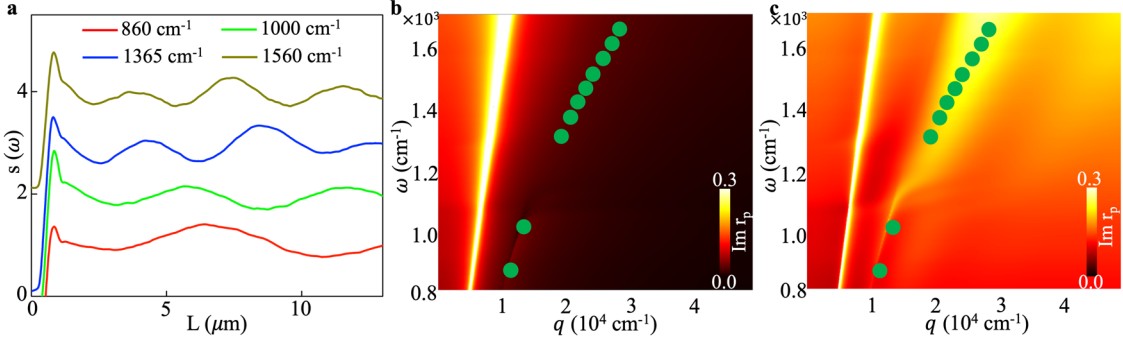

**Fig. 2 | Plasmons on a 80 nm-thick flake of CsV₃Sb₅ and their frequency-momentum dispersion. a** Line-profiles of the extracted near-field scattering amplitude taken along the dashed line in Fig. 1b–e for the same flake at different laser frequencies. **b** The imaginary part of the p-polarized reflection coefficient Im[$r_p$] of the flake calculated with the in-plane component of the dielectric function $\epsilon_{ab}$ obtained from far-field measurements and $\epsilon_c$ from DFT calculations. The dots are experimental dispersion data. **c** Same as panel **b** but with $\epsilon_c = 0.6$.

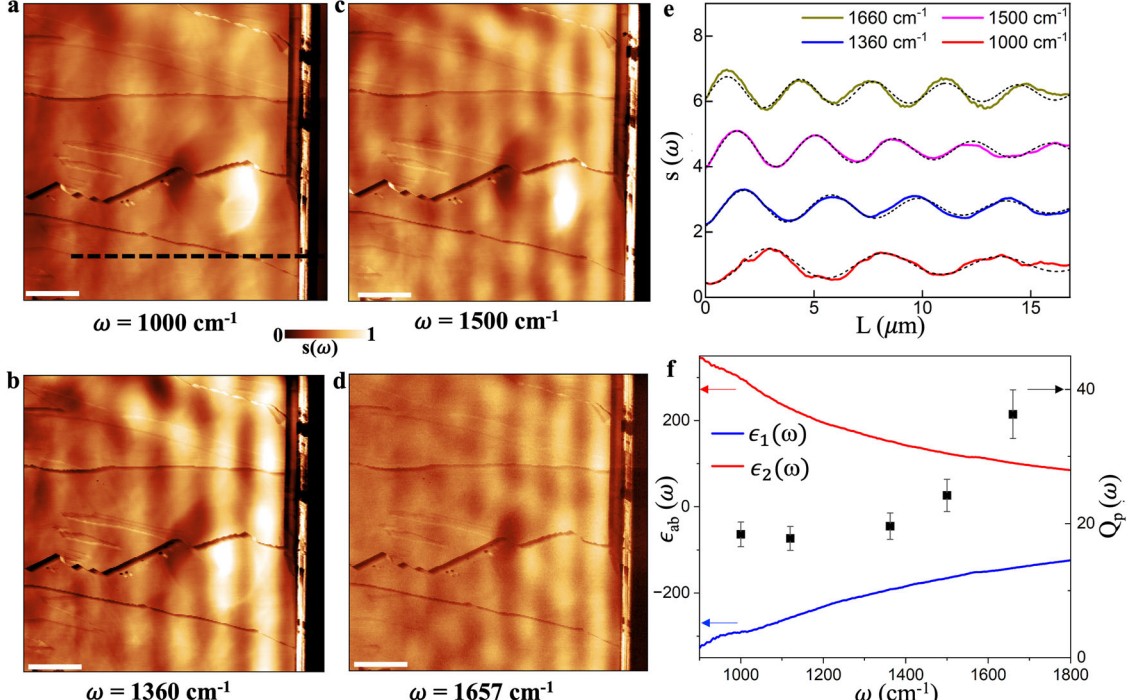

**Fig. 3 | Plasmons on a 380-nm-thick flake of CsV₃Sb₅. a–d** Images of the near-field amplitude $s(\omega)$ at four different frequencies on a 380-nm-thick crystal. The scale bar is 5 μm in length. **e** Line-profiles of the extracted near-field scattering amplitude at these frequencies. Dashed lines representing fits using the formula described in the main text. **f** The in-plane dielectric function $\epsilon_{ab}(\omega)$ of CsV₃Sb₅ obtained from far-field optical reflectivity measurements[43] and the frequency-dependent quality factors $Q_p(\omega)$ of the plasmons obtained from the line profiles. The error bars represent the 90% confidence intervals.

branches (symmetric and antisymmetric branches between the top and bottom surfaces) in close vicinity of the light cone (see Supplementary Note 4). This is not consistent with our experimental observations, suggesting that the actual $\epsilon_c(\omega)$ differs significantly from the DFT predictions. In fact, we discovered that the Im[$r_p$] computed using a positive $\epsilon_c$ ($\epsilon_c(\omega) = 0.6$) agrees well with the experimental data (Fig. 2c). These findings provide preliminary evidence suggesting that the observed plasmons correspond to hyperbolic modes. In this scenario characterized by $\epsilon_{ab}\epsilon_c < 0$, hyperbolic plasmon polaritons propagate within the bulk of the crystal, featuring an IFS that is an open hyperbola in momentum space, in contrast to the closed sphere or ellipse associated with conventional photons.

## Thickness-dependence of CsV₃Sb₅ plasmon polaritons

We also examined flakes with different thicknesses and found a clear thickness dependence of the plasmon dispersion. In Fig. 3, we show

nano-IR imaging data obtained from a 380-nm-thick crystal. Compared to the 80 nm flakes, the pronounced plasmonic patterns with larger periodicity persist over the entire field of view (Fig. 3a–d). To quantify the spatial decay rate $q_p''$, we fit the averaged near-field line-profiles (Fig. 3e) with the formula $S(x) = A\sin(k^* \cdot x + B)e^{-q_p''x}$ where $k^*$, $q_p''$, A and B are fitting parameters[50,51] (see Supplementary Note 3). The frequency-dependent quality factors, $Q_p(\omega) = q_p/q_p''$, are shown in Fig. 3f, with values as high as 40 at 1650 cm⁻¹, representing some of the highest quality factors for plasmons in layered 2D metals at room temperature.

At frequency $\omega$, the complex momenta of hyperbolic plasmons can be approximated by

$$q_N^2 = -\frac{\epsilon_c}{\epsilon_{ab}}\left(\frac{N\pi}{d}\right)^2 + \epsilon_c k_0^2 \tag{1}$$

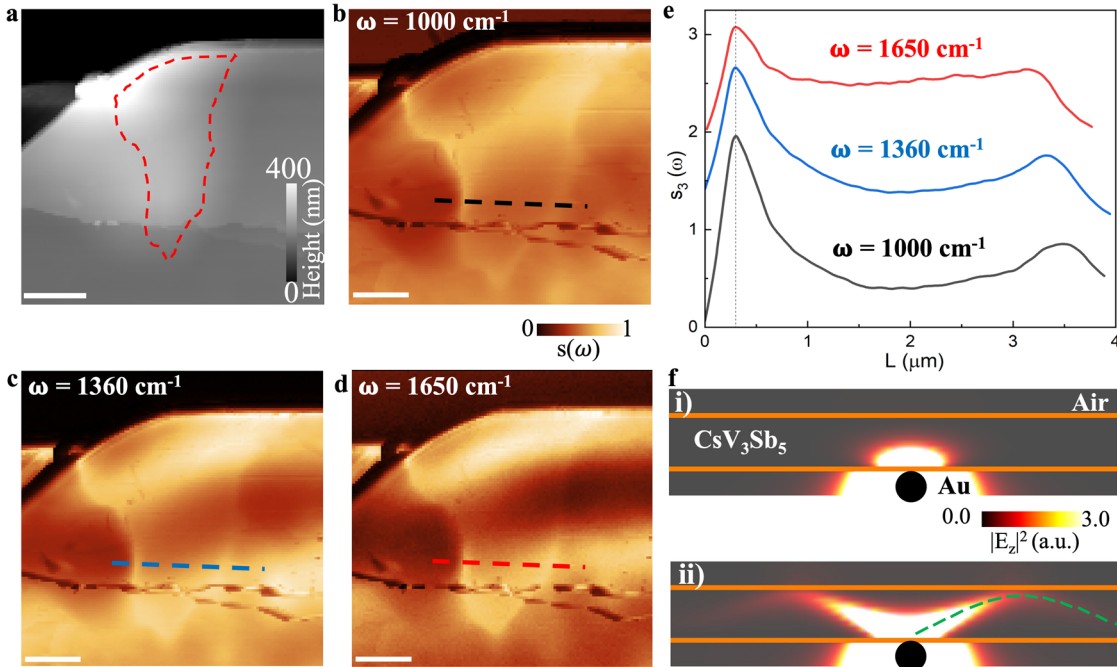

**Fig. 4 | Hyperbolic rays in the Kagome metal CsV₃Sb₅. a** AFM and **b−d** Near-field amplitude images of a 407-nm-thick crystal. A pre-patterned gold launcher underlying the crystal can be clearly visualized through near-field imaging. The scale bar is 2 μm in length. **e** The profile of the near-field amplitude extracted along the dashed lines at the same locations in panels **b−d**. **f** The simulated electric field distribution emitted by a dipole launcher is plotted on a cross-section of the CsV₃Sb₅ flake. For better illustration, panel i) uses $\epsilon_\perp = \mathrm{Re}[\epsilon_{ab}] + \frac{i}{2}\mathrm{Im}[\epsilon_{ab}]$ with $\epsilon_c = -10$, while panel ii) uses the same $\epsilon_\perp$ but with $\epsilon_c = 10$.

for $N \geq 1$ where $N$ is the index of the branch, $d$ is the thickness of the slab, and $k_0 = \omega/c$ is the wave-vector of vacuum photons. The $N = 0$ branch is beyond the description of Eq. (1) and is almost on the vacuum light cone, not consistent with the observed modes. Therefore, we assume that the observed plasmons belong to the $N = 1$ branch since $N > 1$ branches have lower spectra weights in the near-field response. Equation (1) predicts that for thick slabs where $d \gg \frac{\lambda_0}{\sqrt{4|\epsilon_{ab}|}} \sim 300\mathrm{nm}$, the second term dominates, rendering the plasmons almost immune to the large dissipation of the in-plane response $\epsilon_{ab}$ (note that $\mathrm{Im}[\epsilon_{ab}]/\mathrm{Re}[\epsilon_{ab}] \sim 1$ in the relevant frequency range, see Fig. 3f). Physically, this is because the electric field of these modes is mostly in the out-of-plane direction (see Supplementary Note 4). Accordingly, Fig. 3 reveals plasmons with high-quality factors in a 380 nm thick crystal. As another evidence of the hyperbolicity, higher-order ($N = 2$) modes in Eq. (1) are observed in thinner CsV₃Sb₅ flakes. In a 17 nm thick crystal, multiple fringes with distinct periodicities appear close to the edge (see Supplementary Note 2). The extracted plasmon momenta match the $N = 1$ and $N = 2$ branches of the calculated dispersion. Based on Eq. (1), these plasmons shall have larger damping rates since their corresponding electric fields have substantial in-plane components. Therefore, only a few fringes were detected in the vicinity of the sample edge. It is also worth noting that the wavelength of the observed $N = 2$ mode reaches ~250 nm, corresponding to a confinement ratio of ~30, similar to previously studied hyperbolic phonon polaritons[23,24,28].

## Hyperbolic rays in the Kagome metal CsV₃Sb₅

We now present another evidence for the hyperbolic nature of these plasmons by directly imaging the hyperbolic rays emitted by a micro-scale gold launcher, as shown in Fig. 4. We transferred a 407-nm-thick CsV₃Sb₅ flake on top of a 50 nm thick predefined gold pattern. In the AFM topography image (Fig. 4a), the prepatterned gold is entirely masked by the thick CsV₃Sb₅ layer. In striking contrast, the near-field images clearly resolve the prepatterned gold structure with bright

contrast and sharp edges (Fig. 4b−e). If $\epsilon_c$ were negative as predicted by DFT, the surface plasmons launched by the tip/gold launcher would decay exponentially away from the surface, with a decay length of $l_{\mathrm{decay}} \approx l_d \sqrt{\frac{\epsilon_c}{\epsilon_{ab}}} \sim 5\,\mathrm{nm}$. Here, $l_d \sim 50\,\mathrm{nm}$ represents the typical wavelength of the electric field launched by the tip. Considering that $\epsilon_{ab} \sim 100 - 200$ and $|\epsilon_c| \sim 1$, we assumed $\sqrt{\frac{\epsilon_c}{\epsilon_{ab}}} \sim 1/10$. Therefore, it would not be possible to observe the gold launcher with such sharp edges underlying a 407-nm-thick slab through normal surface plasmons (Fig. 4f(i))[52]. From this observation, we conclude that $\epsilon_c$ must be positive, such that the signal carriers are hyperbolic plasmonic rays that propagate through the bulk at a propagation angle θ (with respect to c-axis) set by $\tan(\theta) = \frac{i\sqrt{\epsilon_{ab}(\omega)}}{\sqrt{\epsilon_c(\omega)}}$, as shown by the simulation results in Fig. 4f(ii) and Supplementary Note 4. We note that for practical hyperlensing applications, an involved reconstruction process is required[53].

## Out-of-plane dielectric function $\epsilon_c(\omega)$ of CsV₃Sb₅

The dependence of the plasmon wavelength on the flake thickness $d$ offers further insight into the system. As shown in Fig. 5a, the increment in the plasmonic wavelength with $d$ is much slower compared to both the surface plasmons and the $N = 0$ branch of the hyperbolic plasmons. Instead, it qualitatively aligns with the predictions of the $N = 1$ hyperbolic branch, which further supports our earlier assumptions. Employing Eq. (1), we extract the $\epsilon_c$ from the measured complex plasmon momenta for the first time, as illustrated in Fig. 5b. From the measured complex plasmon momenta $q_N$ for $N = 1$ and the experimentally obtained in-plane dielectric function $\epsilon_{ab}(\omega)$ by far-field measurements, we invert Eq. (1) to extract the $\epsilon_c$ which is shown in Fig. 5b. While $\epsilon_{ab}(\omega)$ can be qualitatively captured by DFT[43], our extracted out-of-plane dielectric function exhibits two significant deviations from the DFT predictions. Firstly, contrary to DFT, the extracted $\mathrm{Re}[\epsilon_c]$ *is* positive within the relevant IR frequency range, transforming the

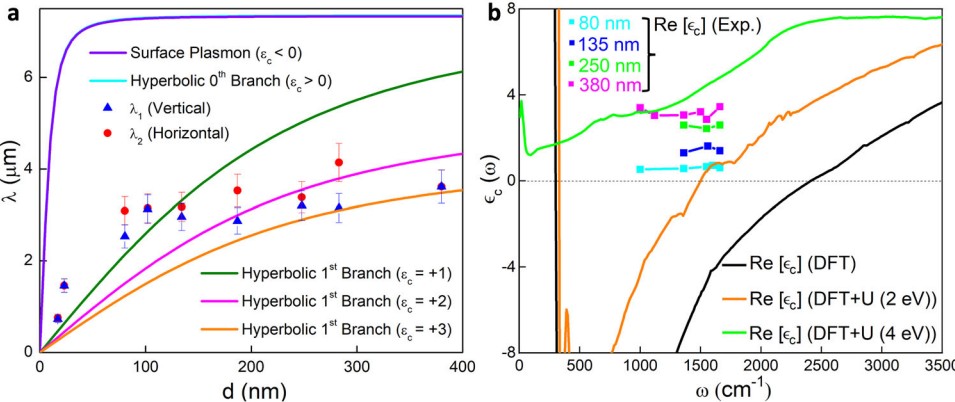

**Fig. 5 | Hyperbolicity in the Kagome metal CsV₃Sb₅. a** Plasmon wavelengths as functions of the flake thickness d at the frequency $\omega = 1360$ cm$^{-1}$. The red dots/blue triangles are experimental plasmon wavelengths extracted from fringes moving in the horizontal/vertical directions, respectively. The error bars represent the 90% confidence intervals. The solid lines are the theoretical predictions using several different $\epsilon_c$ (see Supplementary Note 4). **b** The out-of-plane dielectric function $\epsilon_c(\omega)$ of CsV₃Sb₅ plotted versus frequency. The solid black, orange and green curves are produced from DFT and DFT + U calculations (with the U value given in brackets), while the magenta, green, blue and cyan-colored dots signify the extracted experimental data.

surface plasmons into the hyperbolic bulk plasmons. Secondly, Im[$\epsilon_c$] is considerably lower than the DFT result from inter-band transitions, leading to a substantial reduction in plasmonic losses. This is consistent with the high-quality factors and propagating plasmon fringes observed in the nano-IR imaging data presented in Figs. 1 and 3. We note that the DFT curve in Fig. 5b includes only the inter-band part of the dielectric function, omitting the Drude part (intra-band). Incorporating the Drude part would further shift the predicted Re[$\epsilon_c$] towards the negative direction. The dramatic discrepancy between our experimental results and the DFT predictions of $\epsilon_c(\omega)$ is likely due to electronic correlations, as detailed below.

## Discussion

We now discuss the possible mechanisms that could cause a sign change of $\epsilon_c(\omega)$ over the relevant frequency range that leads to hyperbolicity. One potential explanation is the presence of correlation-induced reorganization of the vanadium bands when on-site Coulomb interaction is considered. To investigate the possible role of correlations, we performed systematic DFT + U calculations by including an on-site Coulomb interaction U. Our results indicate that varying U from 0 to 4 eV leads to a significant change of the out-of-plane optical response (Fig. 5b). Without on-site U, the real part of the c-axis optical conductivity (see Supplementary Note 5) exhibits an inter-band absorption peak around 400 cm$^{-1}$. However, when U = 4 eV, this peak is completely suppressed due to the shifting of the electronic bands composed of vanadium d-orbitals. As a result, a substantial portion of the spectral weight is transferred to higher energy beyond 2000 cm$^{-1}$. From the Kramers-Kronig relation, an absorption peak tends to contribute a negative Re[$\epsilon_c$] on its high-frequency side and a positive Re[$\epsilon_c$] on its low-frequency side. Therefore, as U is tuned from 0 to 4 eV, the corresponding Re[$\epsilon_c$] undergoes a sign change from negative to positive values over the mid-IR and far-IR frequency ranges, which qualitatively aligns with our experimental results (Fig. 5b).

Finally, the observation of tunable, low-loss hyperbolic plasmons across a wide IR frequency range benefits from the unique combination of features in Kagome metals, including a high density of itinerant carriers, a quasi-2D structure, and electronic correlations. We demonstrated the feasibility of deriving the out-of-plane dielectric function $\epsilon_c(\omega)$ in CsV₃Sb₅ via employing the evanescent near-field detection, which has also been utilized in biaxial α-MoO₃ to extract its dielectric function[54]. Unlike the extensively studied hyperbolic phonon polaritons, our findings represent the first occurrence of possible hyperbolic plasmons in a natural crystal facilitated by electronic correlations. The higher-order hyperbolic branches with enhanced confinement reported here constitute a new way to explore the potential for waveguiding and nanoscale light focusing through the hyperbolic plasmonic channel[55]. Furthermore, as depicted in Fig. 5b, $\epsilon_c(\omega)$ is likely to maintain a positive trend at higher frequencies, extending beyond our current detection range while $\epsilon_{ab}$ remains negative below the plasma frequency of about 8000 cm$^{-1}$, indicating a much broader spectrum range of hyperbolicity in CsV₃Sb₅. We anticipate similar effects in the closely related compounds such as KV₃Sb₅[56] and other Kagome metals, and envision that electronic correlations and dimensional confinement in these compounds may open up new avenues for engineering the properties of plasmon polaritons in the technologically important mid-IR to far-IR frequency range.

## Methods

### Sample synthesis and device fabrications

The single crystals of CsV₃Sb₅ were grown from Cs ingot (purity 99.9%), V powder (purity 99.9%), and Sb grains (purity 99.999%) using the self-flux method inside an argon glovebox with oxygen and moisture levels <0.5 ppm, as described previously[34]. The crystallinity of CsV₃Sb₅ has been examined via X-ray diffraction (XRD) measurement[34]. Briefly, we mechanically exfoliated the as-grown bulk crystals onto Si/SiO₂ substrate and pre-inspected the quality of the CsV₃Sb₅ flakes via an optical microscope and AFM inside a purged nitrogen environment. The AFM topography image of CsV₃Sb₅ flakes is shown in Supplementary Note 1. For pre-patterned Au antenna devices, Au/Cr (50 nm/3 nm) patterns were defined using the standard e-beam lithography on SiO₂/Si substrates. Thin CsV₃Sb₅ flakes were then directly exfoliated on pre-patterned Au structures.

### Nano-infrared measurements

The infrared nano-imaging experiments were performed using s-SNOM (NeaSpec) equipped with continuous wave mid-IR quantum cascade lasers. The s-SNOM is based on AFM with curvature radius ~20 nm operating in the tapping mode with a tapping frequency around 270 kHz. A pseudo-heterodyne interferometric detection module was implemented to extract both the scattering amplitude *s* and the phase of the near-field signal. In the current work, we discuss the amplitude of the signal. In order to subtract the background signal, we demodulated the near-field signal at the 3th harmonics of the tapping frequency.

### In-plane optical spectra of CsV₃Sb₅

The in-plane dielectric function $\epsilon_{ab}(\omega)$ of CsV₃Sb₅ in mid-IR frequency range was obtained through Kramers-Kronig analysis of the measured

in-plane reflectivity spectra. Fourier transform infrared spectroscopy (FTIR) was utilized for the in-plane reflectivity measurements, employing a Vertex80v spectrometer coupled with a Hyperion IR microscope. To establish a reference, freshly evaporated Au was used in the measurements.

## Computational details

The density-functional theory (DFT) was applied to calculate the band-structure of $CsV_3Sb_5$, using Perdew-Burke-Ernzerh exchange-correlation potential (see Supplementary Note 5). Self-consistent calculations and structural relaxations were converged on the $36 \times 36 \times 18$ k-mesh for the $CsV_3Sb_5$ structure. To account for on-site Coulomb interactions on the localized vanadium atoms, the DFT + U approach was employed. As it is impractical to directly measure the out-of-plane dielectric function $\epsilon_c(\omega)$ of $CsV_3Sb_5$ using conventional far-field based FTIR, we computed $\epsilon_c(\omega)$ using both DFT and DFT + U and compare with the experimentally obtained $\epsilon_c(\omega)$ via nano-IR measurements. Furthermore, spin-orbit coupling was included in both of the band structure and optical conductivity calculations.

## Data availability

All data needed to evaluate the conclusions in the paper are present in the paper and/or the Supplementary Materials. Additional data related to this paper may be requested from the authors upon reasonable request.

## Code availability

Code used for the electric field distribution and DFT calculations is available from the corresponding author upon reasonable request.

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

## Acknowledgements

N.G.X. acknowledges discussions with K.Yang, Y.C.Wang, M.X.Ye. Research of scanning near-field nano-optical imaging studies is supported by U.S. Department of Energy (DOE) Early Career Research Program, Office of Science, Basic Energy Sciences (BES), under award DE-SC0022022 (N.G.X.). Research on 2D polaritonics is supported by the National Science Foundation (NSF) CAREER award, under award DMR-2145074 (N.G.X.). N.G.X. acknowledges the support from ACS-DNI (PRF# 66465-DNI10), the Start-Up Fund from Florida State University, and the National High Magnetic Field Laboratory (NHMFL). The NHMFL is supported by the NSF through DMR-1644779 and the state of Florida. Research at Tsinghua University is supported by the startup grant from State Key Laboratory of Low-Dimensional Quantum Physics and Tsinghua University. S.D.W. and B.R.O. gratefully acknowledge support via the UC Santa Barbara NSF Quantum Foundry funded via the Q-AMASE-i program under award DMR-1906325. B.R.O. gratefully acknowledges support from DOE/Office of Science/BES, Materials Sciences and Engineering Division.

## Author contributions

G.X.N. conceived the ideas and designed the experiments. G.X.N, H.S., A.G. and S.C. performed the nanoscale infrared measurements and characterizations. Z.S., G.X.N. H.S., A.G. and B.Y. performed theoretical analysis and modeling of the data. B.R.O. and S.D.W. synthesized the crystal. E.U. helped with data analysis and A.A.T. performed DFT calculations. G.X.N. and Z.S. co-wrote the manuscript with input from all co-authors.

## Competing interests

The authors declare no competing interests.
