## [Peer Review File · Nature Communications]

REVIEWER COMMENTS

Reviewer #1 (Remarks to the Author):

This paper explores broadband hyperbolic plasmons within a Kagome metal CsV₃Sb₅, employing near-field s-SNOM measurements to enable the extraction of the out of plane dielectric permittivity, which allowed the authors to demonstrate that the current DFT-derived out of plane response was not accurate. Overall the paper highlights interesting and novel behavior within an emerging class of materials, while also enabling advancements in our understanding of the optical properties of these systems. As such, in principal I believe the paper is suitable for publication in Nature Communications after the following comments are addressed.

1. The authors state that the out of permittivity is not able to be extracted via far-field techniques. This is confusing as provided the ellipsometric or FTIR measurements at angles beyond the Brewster angle are employed such measurements should be reasonably straightforward. I can understand sometimes this can be challenging for 2D crystals due to the small size, and thus the need to employ an FTIR microscope, but this technique has been used for an array of 2D materials including isotopically enriched hBN, Hf-based TMDs, and MoO₃ among others. Could the authors clarify why this isn't possible within this system?

2. One of the major claims of this work is also the use of the near-field measurements to derive the dielectric function, specifically the out of plane response. However, it should be noted that this has been used previously for MoO₃ (<https://onlinelibrary.wiley.com/doi/abs/10.1002/adma.201908176>). This should be discussed in the context of this work.

3. The authors also claim "However, the hyperbolic plasmon polaritons in naturally occurring layered correlated 3D metals remain largely unexplored, particularly in the long-wavelength infrared (IR) and terahertz (THz) frequency ranges." However, it should be noted that far-IR hyperbolic plasmons have been reported in WTe₂ <https://www.nature.com/articles/s41467-020-15001-9>. Again, these results should be discussed in the context of that work.

4. There are a number of English grammar issues, for instance on page 3 the authors state "enabling the launch of plasmonic waves...". This is one example. Another is on page 10 the use of the 'word' "increasement"? There are several others. I would advise the authors ask a native English speaker to proof-read.

5. The authors discuss the hyperbolic modes in multiple samples in multiple configurations. For instance, in Fig. 1 they look at the modes scattered and reflected by irregular edges, from metal disks, line profiles from thicker flakes in Fig. 3, then some hard to follow hyperlens results in Fig. 4. I think this could be simplified a great deal without losing the key messages.

6. The authors point to the use of s-SNOM to identify that the material is actually hyperbolic, pointing to the out of plane permittivity being 0.6 rather than a negative value as discussed in a

previous DFT paper. However, they never clearly state where that value comes from? I presume it was empirically derived? Could you clarify?

7. Fig. 3f is hard to follow. I would suggest the authors use arrows to show that the black data points refer to the right axis, while the two lines to the left.

8. The authors state that they observe higher order hyperbolic polariton modes in THINNER flakes. Yet for hyperbolic systems the dispersion flattens out with decreasing thickness pushing the higher order branches to much higher wavevectors, which are typically inaccessible by s-SNOM. These modes are usually much easier to observe in thicker samples, so it is curious why this is reversed here. The authors should provide some insight into this effect.

9. I cannot make sense of Fig. 4. This is supposed to be a hyperlens image? If so, is it of some bizarre random metal structure? Why wasn't a traditional shape employed if this is the case? This would have been easier to verify the behaviors. Further, such hyperlensing typically results in multiple reflections giving rise to a field profile that is not a direct replication of the underlying object. It is unclear why something like that is seen here? either way, this figure and these results are not convincing and should be improved and discussed more clearly.

10. The authors point to the fact that their results helped correct the value of the out of plane permittivity, which I agree with. However, they also refer to the results giving improvements in $\text{Im}(\epsilon)$ from the DFT. Yet, DFT doesn't offer clear paths towards calculating losses, only the resonance frequencies. This should be clarified.

11. On page 12 the authors state "From the Kramer-Kronig relation, an absorption peak naturally leads to a negative $\text{Re}[\epsilon_c]$ on its high frequency side and a positive $\text{Re}[\epsilon_c]$ on its low-frequency side." This can be true, for instance in the context of TO phonons from a polar crystal. However, this isn't always the case as in molecular vibrational resonances.

Reviewer #2 (Remarks to the Author):

The authors reported a hyperbolic plasmonic material with extensive experimental confirmation in the mid-infrared region. The material system is intriguing and it is of broad interest, and the experimental data are convincing with a lot of details. I recommend the publication in Nature communications if the following technical questions can be addressed:

1. Wording. Line 36: "natural hyperbolic materials can sustain a much broader range of hyperbolic frequencies and momenta, allowing them to accommodate the aforementioned phenomena without the need for extensive fabrication processes". This is misleading because: (1) the frequency of natural hyperbolic materials is rather limited, at least not an advantage when compared with artificial materials. (2) the momenta range is due to the loss of the system, and poor natural hyperbolic material could be worse than high-quality artificial hyperbolic materials.

2. Quality factor. The definition of quality factor in different references varies greatly regarding polariton propagation. It is recommended that the Q-factor is compared in the context of other well-known materials, such as hBN and/or MoO₃, so that readers have a better understanding of the polaritons: are they better/worse/comparable to something they already know?

3. Hyperlens data. I have two concerns about the hyperlens data:

a. The frequency dependence is relatively weak. Does the data agree with simulations quantitatively?

b. For hyperlens imaging, it typically leads to a ring for every single dipole (also shown in figure 4f). Therefore, the image on the top surface should have two edges rather than a direct duplicate of the object (see reference Li, Peining, et al. "Hyperbolic phonon-polaritons in boron nitride for near-field optical imaging and focusing." Nature communications 6.1 (2015): 7507.). Why did the authors not observe the "double-edged" image?

Reviewer #3 (Remarks to the Author):

This work reports infrared (IR) nano-imaging of thin flakes of CsV₃Sb₅, where they observe propagating plasmon waves in real-space with wavelengths tunable by the flake thickness. Moreover, the plasmon is believed to be hyperbolic bulk plasmons, which shows low dissipation. Overall, this work is well conducted and written. However, some major issues still need to be supplemented. Detailed comments are as followed:

(1) The authors claimed that "collective electronic excitations in this class of Kagome metals remain completely unknown." Actually, the recent work by Ma et. al have reported the surface plasmon properties of KV₃Sb₅ (J. Mater. Chem. C, 2022,10, 18393-18403)

(2) As described, CsV₃Sb₅ microcrystals were obtained using mechanical exfoliation. This means that the interatomic layer interaction is quite weak. Moreover, the V atoms form in-plane layers. However, the authors claimed that the electronic correlations between V atoms are more pronounced along the out-of-plane direction, which leads to the positive real part dielectric function. This is not understandable.

(3) The reason for the thickness-dependent plasmon properties of the films should be given.

(4) The purity and crystallinity of the CsV₃Sb₅ microcrystals should be demonstrated to verify that the hyperbolic plasmon is intrinsic property.

(5) In the SI, R_p is regarded as the reflection coefficient, and r_p is also regarded as reflection coefficient, what is the difference between them.

(6) On page 7 of SI, the authors claimed that the condition for the plasmon modes of the slab refers to S3. However, there is not such condition in the S3. The detailed derivation of the condition should be provided. Moreover, the detailed derivation for the surface plasmon and hyperbolic plasmon dispersion condition should be provided, or relevant reference should be given.

(7) On page 10 of SI, the authors claimed that the dispersion condition for the hyperbolic plasmon modes is obtained from Eq. (1). However, there is not equation labelled Eq. (1).

Reply to Reviewer 1:

This paper explores broadband hyperbolic plasmons within a Kagome metal CsV_3Sb_5 , employing near-field s-SNOM measurements to enable the extraction of the out of plane dielectric permittivity, which allowed the authors to demonstrate that the current DFT-derived out-of-plane response was not accurate. Overall the paper highlights interesting and novel behavior within an emerging class of materials, while also enabling advancements in our understanding of the optical properties of these systems. As such, in principal I believe the paper is suitable for publication in Nature Communications after the following comments are addressed.

Our general response: We thank the reviewer for recognizing the novelty of our work.

1.1. The authors state that the out-of-permittivity is not able to be extracted via far-field techniques. This is confusing as provided the ellipsometric or FTIR measurements at angles beyond the Brewster angle are employed such measurements should be reasonably straightforward. I can understand sometimes this can be challenging for 2D crystals due to the small size, and thus the need to employ an FTIR microscope, but this technique has been used for an array of 2D materials including isotopically enriched hBN, Hf-based TMDs, and MoO_3 among others. Could the authors clarify why this isn't possible within this system?

Response: We thank the reviewer for the critical reading and appreciate the reviewer's concise summary of FTIR techniques.

We agree with the reviewer that the FTIR microscope is adept at scrutinizing 2D materials close to the diffraction limit. The as-grown CsV_3Sb_5 crystal, with commendable lateral dimension in the ab-plane, faces challenges due to its thickness along the c-direction approaches the diffraction limit at our relevant middle/far-infrared frequencies. Additionally, the irregular and rough edges along the c-direction, coupled with the crystal's softness, pose practical difficulties in employing FTIR measurements to unveil the out-of-plane permittivity of CsV_3Sb_5 .

To eliminate any potential confusion, we have rephrased the previous sentence in the updated version. Instead of saying 'Out-of-plane permittivity is not able to be extracted via far-field techniques', in the revised version, we say "Out-of-plane permittivity is generally difficult to be extracted via conventional far-field techniques."

1.2. One of the major claims of this work is also the use of the near-field measurements to derive the dielectric function, specifically the out-of-plane response. However, it should be noted that this has been used previously for MoO_3 (<https://onlinelibrary.wiley.com/doi/abs/10.1002/adma.201908176>). This should be discussed in the context of this work.

Response: We thank the reviewer for pointing this out. In the revised version, we have incorporated a discussion and specifically cited this work in ref. 54 to acknowledge their pioneering contribution as follows: “We demonstrated the feasibility of deriving the out-of-plane dielectric function $\epsilon_c(\omega)$ in CsV₃Sb₅ via employing the evanescent near-field detection, which has also been utilized in biaxial α -MoO₃ to extract its dielectric function⁵⁴.”

1.3. The authors also claim "However, the hyperbolic plasmon polaritons in naturally occurring layered correlated 3D metals remain largely unexplored, particularly in the long-wavelength infrared (IR) and terahertz (THz) frequency ranges." However, it should be noted that far-IR hyperbolic plasmons have been reported in WTe₂ <https://www.nature.com/articles/s41467-020-15001-9>. Again, these results should be discussed in the context of that work.

Response: We thank the reviewer for the constructive feedback. In the revised version, we have explicitly cited this pioneering work as ref. 30 in the following: “Systematic investigations have been conducted on natural layered two-dimensional (2D) hyperbolic insulators²⁰ such as hexagonal boron nitride^{21,22,23,24}, MoO₃^{25,26,27}, V₂O₅²⁸, and metals such as ZrSiSe²⁹ and WTe₂³⁰. However, the hyperbolic plasmon polaritons in correlated 2D metals remain largely unexplored, particularly in the long-wavelength infrared (IR) and terahertz (THz) frequency ranges.”

1.4. There are a number of English grammar issues, for instance on page 3 the authors state "enabling the launch of plasmonic waves...". This is one example. Another is on page 10 the use of the 'word' "increasement"? There are several others. I would advise the authors ask a native English speaker to proof-read.

Response: We appreciate the reviewer for the critical reading of our manuscript. In the revised version, we have fixed these and other grammar issues. The revised manuscript reflects these improvements, and we are grateful for the reviewer's diligence in bringing these matters to our attention.

1.5. The authors discuss the hyperbolic modes in multiple samples in multiple configurations. For instance, in Fig. 1 they look at the modes scattered and reflected by irregular edges, from metal disks, line profiles from thicker flakes in Fig. 3, then some hard-to-follow hyperlens results in Fig. 4. I think this could be simplified a great deal without losing the key messages.

Response: We appreciate the reviewer for the constructive suggestion. We have carefully considered how to best present the results coherently. Given the comprehensive nature of our study,

aiming to encapsulate a full picture within a single publication, we prefer to retain the current format. The justification is the following:

We began our exploration with nano-IR probing to show propagating plasmon waves from a thin 80 nm CsV₃Sb₅ flake at variable frequencies, as depicted in Fig. 1. Following this, we conducted a detailed frequency-momentum dispersion analysis in Fig. 2, revealing the first hint of hyperbolic plasmons. Furthermore, in Fig. 3, we present an examination of plasmons on a thicker (380 nm) CsV₃Sb₅ flake, providing additional evidence to underscore the generality of the observed features and elucidate its thickness dependence.

Following the presentation of nano-IR imaging from planar CsV₃Sb₅ flakes, we demonstrate the hyperlensing effect by imaging the buried metal structure underlying CsV₃Sb₅ flakes in Fig. 4. This serves as the second indication of hyperbolic plasmons in this compound. A comprehensive explanation of the observed hyperlensing imaging is provided in question 1.9 below.

Figure 5 is our summary plot, which encompasses both the thickness-dependence studies of CsV₃Sb₅ flakes and a comparison of experimentally extracted out-of-plane permittivity with DFT/DFT+U calculations. From this, we identify electronic correlations as the stimuli for the observed hyperbolic plasmonic characteristics.

We again thank the reviewer for their constructive suggestions, and we hope the reviewer finds our justification reasonable.

1.6. The authors point to the use of s-SNOM to identify that the material is actually hyperbolic, pointing to the out-of-plane permittivity being 0.6 rather than a negative value as discussed in a previous DFT paper. However, they never clearly state where that value comes from? I presume it was empirically derived? Could you clarify?

Response: We thank the reviewer for the careful examination of our work. We used two methods to determine the value of the out-of-plane permittivity. In the first method, we used the imaginary part of the reflection coefficient simulation. Through systematic tuning of the out-of-plane permittivity value as a tuning parameter, we aligned the simulated dispersion with experimentally extracted plasmon dispersion in Fig. 2.

As a sanity check, we have also specifically extracted the value of out-of-plane permittivity, with a second method based on the analytical expression of the plasmon dispersion in Eq. (1), as shown in Fig. 5b. It is worth noting that this method is free of fitting parameters.

In the revised version, we have included a brief discussion to clarify the out-of-plane permittivity value extraction in page 10 as the following: “From the measured complex plasmon momenta q_N for $N = 1$ and the experimentally obtained in-plane dielectric function $\epsilon_{ab}(\omega)$ by far-field measurements, we invert Eq. (1) to extract the ϵ_c which is shown in Fig. 5b. While $\epsilon_{ab}(\omega)$ can be qualitatively captured by DFT⁴³, our extracted out-of-plane dielectric function exhibits two significant deviations from the DFT predictions.”

1.7. Fig. 3f is hard to follow. I would suggest the authors use arrows to show that the black data points refer to the right axis, while the two lines to the left.

Response: We thank the reviewer for his constructive suggestion. In the revised version, we have added the arrows in Fig .3f.

1.8. The authors state that they observe higher-order hyperbolic polariton modes in THINNER flakes. Yet for hyperbolic systems the dispersion flattens out with decreasing thickness pushing the higher-order branches to much higher wavevectors, which are typically inaccessible by s-SNOM. These modes are usually much easier to observe in thicker samples, so it is curious why this is reversed here. The authors should provide some insight into this effect.

Response: We thank the reviewer for his critical reading. We think the potential reason is as follows: Unlike materials like hBN, CsV₃Sb₅ exhibits hyperbolic modes with significantly smaller wave vectors at the same thickness, making them relatively harder to excite by the tip. In thinner CsV₃Sb₅ flakes, higher-order mode appears due to their larger wave vectors compared to thicker ones, thus facilitating more efficient coupling with the tip.

1.9. I cannot make sense of Fig. 4. This is supposed to be a hyperlens image? If so, is it of some bizarre random metal structure? Why wasn't a traditional shape employed if this is the case? This would have been easier to verify the behaviors. Further, such hyperlensing typically results in multiple reflections giving rise to a field profile that is not a direct replication of the underlying object. It is unclear why something like that is seen here? either way, this figure and these results are not convincing and should be improved and discussed more clearly.

Response: We thank the reviewer for the critical reading. We tried different metal structures which showed identical super-resolution imaging behavior. The reason why we did not see multiple reflections is the following:

In previous studies of hyperbolic phonon polaritons, the clarity of hyperlensing benefits from highly dispersive dielectric permittivity and a much lower propagation angle. The propagation

angle θ is given by $\tan \theta = \frac{i\sqrt{\epsilon_{ab}(\omega)}}{\sqrt{\epsilon_c(\omega)}}$ with ϵ_c and ϵ_{ab} representing the out-of-plane and in-plane permittivities, respectively. Taking hBN as an example, the ratio of ϵ_{ab} to ϵ_c varies considerably in the upper Reststrahlen band (frequency between 1360 cm^{-1} and 1610 cm^{-1}). As a result, the propagation angle θ varies from $\sim 83^\circ$ to 2° in the type-II upper Reststrahlen band. However, to illustrate the multiple reflection events, one typically adjusts the incident light frequency within the range of $1400\text{-}1530 \text{ cm}^{-1}$, which corresponds to tuning the propagation angle θ between $45^\circ\text{-}60^\circ$. This range provides the optimal conditions for visualizing a well-resolved double-edged ring pattern emanating from the underlying Au structure. This is also clearly noted in one of the pioneering works on the first observation of hyperbolicity in hBN by Li *et al.* *Nature Communications*, 6, 7507 (2015): as incident light wavelengths with very low/high propagation angles, the image of the sub-diffraction Au pattern is nearly perfectly restored on the top surface of the hBN, similar to near-field superlensing (page 3, second paragraph in Li *et al.* *Nature Communications* (2015)).

In our case, the ϵ_{ab} to ϵ_c ratio is large across the entire detectable mid-IR frequency regime, such that the propagation angle θ remains large ($> 80^\circ$). Together with the high loss of the correlated metal, the hyperbolic ray has decayed significantly after one reflection from the top surface, as shown by the schematic numerical simulation in Fig. 4ii. Due to this large propagation angle θ , we only see the replication of the outer profile of the underlying structure.

However, it is important to note that we observed a very clear, consistent variation of the size of the outer profile as a function of the incident light frequency, as depicted in Fig. 4. This frequency dependence of the observed near-field imaging strongly indicates its hyperlensing characters.

1.10. The authors point to the fact that their results helped correct the value of the out-of-plane permittivity, which I agree with. However, they also refer to the results giving improvements in $\text{Im}(\epsilon)$ from the DFT. Yet, DFT doesn't offer clear paths towards calculating losses, only the resonance frequencies. This should be clarified.

Response: We thank the reviewer for pointing this out. Regarding $\text{Im}(\epsilon)$ or loss in the optical conductivity (or equivalently, the permittivity), both intra-band losses (e.g., Drude loss) and inter-band losses (e.g., inter-band transitions) can contribute significantly. Notably, the latter does not rely on impurities, phonons, or electronic interactions, making it accessible for calculation through DFT.

In our focused middle-IR frequency range which inter-band dominates, DFT predicts substantial inter-band losses, and our nano-IR measurements provide valuable corrections to this prediction. We have clarified in the revised manuscript that $\text{Im}(\epsilon)$ from DFT specifically refers to inter-band transitions.

1.11. On page 12 the authors state "From the Kramer-Kronig relation, an absorption peak naturally leads to a negative $\text{Re}[\epsilon_c]$ on its high-frequency side and a positive $\text{Re}[\epsilon_c]$ on its low-frequency side." This can be true, for instance in the context of TO phonons from a polar crystal. However, this isn't always the case as in molecular vibrational resonances.

Response: We appreciate the valuable feedback from the reviewer. What we aim to convey is that the $\text{Re}[\epsilon]$ arising from an absorption peak has such behavior, and it may not necessarily reflect the net $\text{Re}[\epsilon]$. In certain molecular vibrational resonances, the absorption peak might be too weak, such that its contribution to the $\text{Re}[\epsilon]$ may not surpass the positive background of $\text{Re}[\epsilon]$.

In the revised version, we have modified this sentence to enhance clarity: "From the Kramers-Kronig relation, an absorption peak tends to contribute a negative $\text{Re}[\epsilon_c]$ on its high-frequency side and a positive $\text{Re}[\epsilon_c]$ on its low-frequency side."

Reply to Reviewer 2:

The authors reported a hyperbolic plasmonic material with extensive experimental confirmation in the mid-infrared region. The material system is intriguing, and it is of broad interest, and the experimental data are convincing with a lot of details. I recommend the publication in Nature Communications if the following technical questions can be addressed:

Our general response: We thank the reviewer for the positive comments and feedback. Below we address all questions of the reviewer one by one.

2.1. Wording. Line 36: “Natural hyperbolic materials can sustain a much broader range of hyperbolic frequencies and momenta, allowing them to accommodate the aforementioned phenomena without the need for extensive fabrication processes”. This is misleading because (1) the frequency of natural hyperbolic materials is rather limited, at least not an advantage when compared with artificial materials. (2) the momenta range is due to the loss of the system, and poor natural hyperbolic material could be worse than high-quality artificial hyperbolic materials.

Response: We thank the reviewer for critical reading. In the revised version, we have rephrased this sentence to avoid any inaccuracy. Specifically, line 36 was modified to “Natural hyperbolic materials can sustain a range of hyperbolic frequencies and momenta, allowing them to accommodate the aforementioned phenomena without the extensive fabrication processes required for artificially engineered hyperbolic metamaterials⁶”.

2.2. Quality factor. The definition of quality factor in different references varies greatly regarding polariton propagation. It is recommended that the Q-factor is compared in the context of other well-known materials, such as hBN and/or MoO₃, so that readers have a better understanding of the polaritons: are they better/worse/comparable to something they already know?

Response: We thank the reviewer for constructive suggestions. We also noticed that the definition of quality factor in different references varies. In our present and previous publications, we have always utilized the same definition, namely, $Q_p(\omega) = q_p/q_p''$, whereas q_p and q_p'' represent the real and imaginary part of the complex wave-vector, as shown in Fig. 3f.

Compared to the phonon polaritons in hBN and MoO₃, the quality factors of plasmons in CsV₃Sb₅ are in the same order in the thick flakes (e.g., Fig. 3f), but significantly smaller in thin flakes (e.g., Fig. 2). We note that since CsV₃Sb₅ is a correlated metal, the observed quality factors are already impressive.

2.3. Hyperlens data. I have two concerns about the hyperlens data:

2.3a. The frequency dependence is relatively weak. Does the data agree with simulations quantitatively?

Response: We agree with the reviewer that the frequency dependence appears weak, which is attributed to the minimal variation in out-of-plane permittivity ϵ_c within our detected frequency range.

A preliminary comparison regarding the propagation angle θ reveals overall agreement between the data and simulations. While an accurate quantitative comparison at this stage is hindered by the lack of precise information on ϵ_c , in-plane permittivity ϵ_{ab} of CsV₃Sb₅ as it approaches the 2D limit, and the exact size of the buried Au structure beneath the flake. For instance, inference of ϵ_c values from polariton dispersion (Figs. 2,3,5) shows variation within 0.5~4 in our detected frequency range, which is likely attributable to the aforementioned uncertainty. We acknowledge the importance of systematically investigating hyperlensing effects based on frequency, geometry, and thickness, along with precisely determining θ — tasks slated for future work. We believe our current experiments will inspire exploration into novel types of hyperbolic plasmonic media in layered structures with correlation effects.

2.3b. For hyperlens imaging, it typically leads to a ring for every single dipole (also shown in Figure 4f). Therefore, the image on the top surface should have two edges rather than a direct duplicate of the object (see reference Li, Peining, et al. “Hyperbolic phonon-polaritons in boron nitride for near-field optical imaging and focusing.” Nature Communications 6.1 (2015): 7507.). Why did the authors not observe the “double-edged” image?

Response: We thank the reviewer for this critical question. The reason why we did not see the double-edged ring pattern lies its large propagation angle θ , where $\tan(\theta)$ is given by $\frac{i\sqrt{\epsilon_{ab}(\omega)}}{\sqrt{\epsilon_c(\omega)}}$, with ϵ_c and ϵ_{ab} representing the out-of-plane and in-plane permittivity, respectively. In our case, the ratio of ϵ_{ab} to ϵ_c remains substantial, ranging from 100-200 across the detectable mid-IR frequency regime. Consequently, the propagation angle θ remains $> 80^\circ$ which makes it nearly impossible to discern the double-edged pattern. As a result, our detected super-resolution images exhibit a replication of the underlying Au pattern with a single edge. It is worth noting, however, that we did observe a clear and consistent variation of the size of the outer profile as a function of the incident light frequency, as depicted in Fig. 4.

In the case of hBN, the dielectric permittivity is highly dispersive. The ratio of ϵ_{ab} to ϵ_c varies considerably, providing a much wider tuning range for the propagation angle θ , from $\sim 83^\circ$ to 2°

in the type-II upper Reststrahlen band (corresponding ω changes from 1360 cm^{-1} to 1610 cm^{-1}). As illustrated in the pioneering work done by Li and co-workers, they tuned the incident light frequency to $\sim 1400\text{-}1527\text{ cm}^{-1}$ within the type-II band. Such that the corresponding propagation angle θ falls within the range of $\sim 45^\circ\text{-}60^\circ$, which is the optimal condition for visualizing a well-resolved double-edged ring pattern from the underlying Au structure. This is also clearly noted by Li and co-workers in their work: “as incident light wavelengths with very low/high propagation angles, the image of the sub-diffraction Au pattern is nearly perfectly restored on the top surface of the hBN, similar to near-field superlensing (page 3, second paragraph in Li *et al. Nature Communications* (2015)).”

Reply to Reviewer 3:

This work reports infrared (IR) nano-imaging of thin flakes of CsV_3Sb_5 , where they observe propagating plasmon waves in real-space with wavelengths tunable by the flake thickness. Moreover, the plasmon is believed to be hyperbolic bulk plasmons, which show low dissipation. Overall, this work is well conducted and written. However, some major issues still need to be supplemented. Detailed comments are as followed:

Our general response: We thank the reviewer for the positive feedback on our work and constructive suggestions. In the revised version, we have carefully addressed and incorporated these valuable points.

3.1. The authors claimed that “collective electronic excitations in this class of Kagome metals remain completely unknown.” Actually, the recent work by Ma et. al have reported the surface plasmon properties of KV_3Sb_5 (J. Mater. Chem. C, 2022,10, 18393-18403)

Response: We thank the reviewer for reminding us of this important work. In the revised version, we have specifically cited this work on surface plasmon studies (Ref. 55) to acknowledge their pioneering contribution. We have also revised our claim to “collective electronic excitations in this class of Kagome metals remain largely unknown”.

3.2. As described, CsV_3Sb_5 microcrystals were obtained using mechanical exfoliation. This means that the interatomic layer interaction is quite weak. Moreover, the V atoms form in-plane layers. However, the authors claimed that the electronic correlations between V atoms are more pronounced along the out-of-plane direction, which leads to the positive real part dielectric function. This is not understandable.

Response: We thank the reviewer for critical reading. In our option, there is no contradiction because weak interlayer couplings imply a smaller band dispersion along the out-of-plane direction. Consequently, narrow bands are thus more sensitive to correlations.

Additionally, recent studies have shown the impact of local electron-electron correlation, primarily driven within intra-atomic layers, is relatively weak in the normal state and not sufficient to account for the observed rich physical effects experimentally. Conversely, non-local electronic correlations are believed to play an important role in elucidating the origin of exotic phenomena observed in Kagome AV_3Sb_5 systems.

It should be noted that, at the current stage, the nature of the charge density wave (CDW) and superconductivity in Kagome AV_3Sb_5 systems remains controversial, and there are still open questions that need to be addressed. Some of the recent theoretical and experimental works

suggested that the superconductivity and CDW might be unconventional, which hints at important electron-electron effects. However, a clear understanding of the correlation effects and strengths is still lacking.

Existing theoretical studies mainly focused on the on-site local correlation effects while excluding non-local correlations. Directly probing correlation effects along the out-of-plane direction also poses experimental challenges. Our work indirectly suggests the presence of non-local correlation effects along the out-of-plane direction via the plasmonic channel in the normal state. Future studies, both theoretical and experimental, are highly in need for a comprehensive understanding of the underlying intricate electronic correlation effects in Kagome AV_3Sb_5 systems.

3.3. The reason for the thickness-dependent plasmon properties of the films should be given.

Response: We thank the reviewer for the constructive suggestion. For surface plasmons, when the films are thin enough, the plasmons from the top and bottom surfaces coupled to each other with a coupling strength dependent on the thickness d . Furthermore, the hyperbolic plasmons propagate in the bulk, making the thickness-dependence more pronounced. As shown by Eq. (1) of the manuscript, at frequency ω , the complex momenta of hyperbolic plasmons can be approximated by

$$q_N^2 = -\frac{\epsilon_c}{\epsilon_{ab}} \left(\frac{N\pi}{d}\right)^2 + \epsilon_c k_0^2 \quad (1)$$

where the thickness d enters the dispersion and yields thickness-dependent plasmon properties.

In the revised version, we have added a physical explanation to the text below Eq. (4) of the revised SI.

3.4. The purity and crystallinity of the CsV_3Sb_5 microcrystals should be demonstrated to verify that the hyperbolic plasmon is an intrinsic property.

Response: We thank the reviewer for this constructive suggestion. The single crystals of CsV_3Sb_5 were grown from Cs ingot (purity 99.9%), V powder (purity 99.9%), and Sb grains (purity 99.999%) using the self-flux method inside an argon glovebox with oxygen and moisture levels <0.5 ppm, as detailed in ref. 34. The high purity of the CsV_3Sb_5 crystal is manifested in the obtained X-ray diffraction (XRD) measurement shown in RFig. 1. Briefly, we quantified and measured the crystallinity on CsV_3Sb_5 bulk crystal at 290 K, which is quite high and confirms its high quality.

RFig 1. X-ray diffraction of CsV₃Sb₅ and its fitting at room temperature by using an X-ray diffractometer (D8 Advance, Bruker), with its space group and determined lattice parameters accordingly.

For the CsV₃Sb₅ microcrystals obtained using mechanical exfoliation from bulk crystal, the typical flake size is less than 30 μm, which is way below the minimum crystal size of 100 μm requested for XRD measurements. Despite this, owing to the van der Waals layered crystal nature of CsV₃Sb₅, it is almost certain that the examined microcrystals correspond to the (001) plane, which has been determined as the largest surface in XRD measurements of bulk crystals.

Moreover, the observation of plasmon polariton waves with high-quality factors $Q_p > 20$ serves as another evidence of the high-quality crystal. Compared to the phonon polaritons in polar insulators, i.e., hBN and MoO₃, the quality factors of plasmons in CsV₃Sb₅ are in the same order (e.g., Fig. 3f). We note that since CsV₃Sb₅ is a correlated metal, the observed quality factors are already impressive. In the revised version, we have incorporated a discussion on the purity and crystallinity information of CsV₃Sb₅ within the Methods section of the manuscript.

3.5. In the SI, R_p is regarded as the reflection coefficient, and r_p is also regarded as reflection coefficient, what is the difference between them.

Response: We thank the reviewer for pointing out these ambiguities, both R_p and r_p represent the same reflection coefficient. In the revised version, we have fixed this typo by changing all the symbols to r_p .

3.6. On page 7 of SI, the authors claimed that the condition for the plasmon modes of the slab refers to S3. However, there is not such condition in the S3. The detailed derivation of the condition should be provided. Moreover, the detailed derivation for the surface plasmon and hyperbolic plasmon dispersion condition should be provided, or relevant reference should be given.

Response: We thank the referee for this suggestion. In the revised SI, we have provided more detailed references: “see Appendix B of Ref. [S16], Eq. 1 of Ref. [S3] and Sec. 4 of its supplemental material”. We have also added more explanations of Eqs. (3) and (4) to improve their readability.

3.7. On page 10 of SI, the authors claimed that the dispersion condition for the hyperbolic plasmon modes is obtained from Eq. (1). However, there is no equation labeled Eq. (1).

Response: We thank the reviewer for pointing out this typo. We meant Eq. (3) of the SI, which is now fixed in the revised SI.

List of updated changes in the manuscript

1. In page 1 abstract paragraph, updated the sentence to be: we infer the out-of-plane dielectric function ϵ_c that is **generally difficult to obtain** in conventional far-field optics...
2. In page 2 introduction first paragraph, updated the sentence to be: Natural hyperbolic materials can **sustain a range of** hyperbolic frequencies and momenta, allowing them to accommodate the aforementioned phenomena without the extensive fabrication processes required for artificially engineered hyperbolic metamaterials⁶.
3. In page 2 introduction second paragraph, updated the sentence to be: Systematic investigations have been conducted on natural layered two-dimensional (2D) hyperbolic insulators²⁰ **such as hexagonal boron nitride^{21,22,23,24}, MoO₃^{25,26,27}, V₂O₅²⁸, and metals such as ZrSiSe²⁹ and WTe₂³⁰.**
4. In page 3 first paragraph, updated the sentence to be: However, collective electronic excitations in this class of Kagome metals remain **largely** unknown.
5. In page 4 first paragraph, updated the sentence to be: plasmon momentum mismatch^{29,33,45,46,47}, **enabling** the launching of plasmonic waves with wavelength $\lambda_p < \lambda_{IR}$. The local electric field of the plasmonic waves gets scattered into the far-field by the sample edge, **facilitating the** direct measurement of the plasmonic response with ~ 20 nm spatial resolution.
6. In page 7, updated the Figure 3 with added arrows.
7. In page 9, the last sentence is updated to be: From the measured complex plasmon momenta q_N for $N = 1$ and the experimentally obtained in-plane dielectric function $\epsilon_{ab}(\omega)$ by far-field measurements, we invert Eq. (1) to extract the ϵ_c which is shown in Fig. 5b. While

$\epsilon_{ab}(\omega)$ can be qualitatively captured by DFT⁴³, our extracted out-of-plane dielectric function exhibits two significant deviations from the DFT predictions.

8. In page 11, first paragraph updated the second last sentence to be: From the Kramers-Kronig relation, an absorption peak **tends to contribute** a negative $\text{Re}[\epsilon_c]$ on its high-frequency side and a positive $\text{Re}[\epsilon_c]$ on its low-frequency side.
9. In page 11, second paragraph, added a new sentence: **We demonstrated the feasibility of deriving the out-of-plane dielectric function $\epsilon_c(\omega)$ in CsV₃Sb₅ via employing the evanescent near-field detection, which has also been utilized in biaxial α -MoO₃ to extract its dielectric function⁵⁴.**
10. In page 12, added ref. 55 in first line: We anticipate similar effects in the closely related compounds such as KV₃Sb₅⁵⁵ and other Kagome metals.
11. In page 12, methods section, added a new sentence to be: **The single crystals of CsV₃Sb₅ were grown from Cs ingot (purity 99.9%), V powder (purity 99.9%), and Sb grains (purity 99.999%) using the self-flux method inside an argon glovebox with oxygen and moisture levels <0.5 ppm, as described previously³⁴. The crystallinity of CsV₃Sb₅ has been examined via X-ray diffraction (XRD) measurement³⁴.**
12. In page 16, added two new references:
 54. Álvarez-Pérez, G. *et al.* Infrared permittivity of the biaxial van der Waals semiconductor α -MoO₃ from near- and far-field correlative studies. *Adv. Mater.* **32**, 1908176 (2020).
 55. He, W., Ma, X., Jiang, J., Wu, X., Zhang, J. Kagome metal KV₃Sb₅: an excellent material for surface plasmon and plasmon-mediated hot carrier applications in the infrared region. *J. Mater. Chem. C*, **10**, 18393-18403 (2022).

13. In page 17, added Ethics declarations section.

REVIEWER COMMENTS

Reviewer #1 (Remarks to the Author):

Overall I appreciate the extensive efforts that the authors have made in addressing the reviewer comments and concerns. I only have two comments. In the intro where the authors state that hyperbolic plasmons have not been widely explored, especially in the far IR, this is written right after referencing WTe₂, which is itself a hyperbolic plasmonic material in the far-IR. I think the authors need to add a sentence noting this fact and highlighting what is different/new about the the Kagome system over that material.

Secondly, in the hyperlensing as in the LI et al paper that they authors cite in their response letter, to say that the image is nearly perfectly replicated is patently false. Indeed, the multiple hot rings that develop result in a complex surface field that requires image reconstruction to derive what the underlying structure being imaged is. This was dicussed in detail in <https://pubs.acs.org/doi/abs/10.1021/acs.nanolett.1c01808>, where just such an image processing approach was developed. This needs be corrected and clarified. I still feel that the hyperlensing imaging provided here is not clear and needs additional clarity.

Reviewer #2 (Remarks to the Author):

The authors have addressed my concerns. I have a final suggestion on the manuscript.

The study is limited to $\sim 1000\text{-}1600\text{ cm}^{-1}$, which is likely due to QCL frequency range. However, it is highly recommended to discuss the in-plane permittivity in a much wider frequency range, which can be easily determined by far-field measurement. As indicated in Fig. 5b, the ϵ_z is likely to be positive in higher frequencies (although cannot be experimentally verified), indicating other hyperbolic bands. Thus, such a discussion will inspire more research in this field: there are no natural hyperbolic materials in the $\sim 2000\text{-}4000\text{ cm}^{-1}$ range.

Reviewer #3 (Remarks to the Author):

This work has been revised according to my comments and can be published in the present form.

Reply to Reviewer 1:

Overall, I appreciate the extensive efforts that the authors have made in addressing the reviewer's comments and concerns. I only have two comments. In the intro where the authors state that hyperbolic plasmons have not been widely explored, especially in the far IR, this is written right after referencing WTe₂, which is itself a hyperbolic plasmonic material in the far-IR. I think the authors need to add a sentence noting this fact and highlighting what is different/new about the Kagome system over that material.

Our general response: We thank the reviewer for this suggestion. In the revised manuscript, to highlight their difference, we have modified the sentences in the second paragraph of our manuscript to “Systematic investigations have been conducted on natural layered two-dimensional (2D) hyperbolic insulators²⁰ such as hexagonal boron nitride^{21,22,23,24}, MoO₃^{25,26,27}, V₂O₅²⁸, and hyperbolic metals such as ZrSiSe²⁹ and WTe₂³⁰. However, the effects of electronic correlations on hyperbolic plasmon polaritons in layered metals remain largely unexplored. In addition, hyperbolic plasmons in correlated metals and superconductors^{5,31,32,33} also offer valuable insights into their many-body physics, which are difficult to examine via conventional optical approaches”.

Secondly, in the hyperlensing as in the LI et al paper that the authors cite in their response letter, to say that the image is nearly perfectly replicated is patently false. Indeed, the multiple hot rings that develop result in a complex surface field that requires image reconstruction to derive what the underlying structure being imaged is. This was discussed in detail in <https://pubs.acs.org/doi/abs/10.1021/acs.nanolett.1c01808>, where just such an image processing approach was developed. This needs be corrected and clarified. I still feel that the hyperlensing imaging provided here is not clear and needs additional clarity.

Response: We thank the reviewer for bringing to our attention the paper on image reconstruction in hyperlensing. We acknowledge the importance of the reconstruction algorithm in deriving the shape of the launcher from the near-field image. However, due to the lack of accurate knowledge of out-of-plane permittivity ϵ_z for CsV₃Sb₅, it is impractical for us to utilize the algorithm to reconstruct the launcher with the extreme clarity demonstrated in M. He et al. Nevertheless, we would like to emphasize that our hyperlensing data qualitatively supports the hyperbolicity of plasmons, as evidenced by our frequency-dependent imaging results shown in Fig. 4 of the manuscript. We leave the quantitative hyperlensing study for future work. In the revised version, we have added the sentence “We note that for practical hyperlensing applications, an involved reconstruction process is required⁵⁶” to clarify the point raised by the reviewer, with specific citation to M. He et al in ref. [56] of our updated manuscript.

Reply to Reviewer 2:

The authors have addressed my concerns. I have a final suggestion on the manuscript. The study is limited to $\sim 1000\text{-}1600\text{ cm}^{-1}$, which is likely due to QCL frequency range. However, it is highly recommended to discuss the in-plane permittivity in a much wider frequency range, which can be easily determined by far-field measurement. As indicated in Fig. 5b, the ϵ_z is likely to be positive in higher frequencies (although cannot be experimentally verified), indicating other hyperbolic bands. Thus, such a discussion will inspire more research in this field: there are no natural hyperbolic materials in the $\sim 2000\text{-}4000\text{ cm}^{-1}$ range.

Our general response: We thank the reviewer for this constructive suggestion. In the revised manuscript, we have incorporated the reviewer's suggestion by adding the following sentence in the last paragraph of the main text: "Furthermore, as depicted in Fig. 5b, $\epsilon_c(\omega)$ is likely to maintain a positive trend at higher frequencies, extending beyond our current detection range while ϵ_{ab} remains negative below the plasma frequency of about 8000 cm^{-1} , indicating a much broader spectrum range of hyperbolicity in CsV_3Sb_5 ".